# Melatonin/Sericin Wound Healing Patches: Implications for Melanoma Therapy

**DOI:** 10.3390/ijms25094858

**Published:** 2024-04-29

**Authors:** Katarzyna Adamiak, Vivian A. Gaida, Jasmin Schäfer, Lina Bosse, Clara Diemer, Russel J. Reiter, Andrzej T. Slominski, Kerstin Steinbrink, Alina Sionkowska, Konrad Kleszczyński

**Affiliations:** 1Department of Biomaterials and Cosmetic Chemistry, Faculty of Chemistry, Nicolaus Copernicus University, Gagarin 7, 87-100 Toruń, Poland; kadamiak@doktorant.umk.pl (K.A.); alinas@umk.pl (A.S.); 2Department of Dermatology, University of Münster, Von-Esmarch-Str. 58, 48149 Münster, Germany; vgaida@uni-muenster.de (V.A.G.); jschaef1@uni-muenster.de (J.S.); lbosse1@uni-muenster.de (L.B.); cdiemer@uni-muenster.de (C.D.); kerstin.steinbrink@ukmuenster.de (K.S.); 3Department of Cell Systems and Anatomy, Long School of Medicine, UT Health, San Antonio, TX 78229, USA; reiter@uthscsa.edu; 4Department of Dermatology, Comprehensive Cancer Center, University of Alabama at Birmingham, Birmingham, AL 35294, USA; aslominski@uabmc.edu; 5Pathology and Laboratory Medicine Service, VA Medical Center, Birmingham, AL 35294, USA

**Keywords:** melatonin, sericin, wound healing, biomaterials, scaffolds, hydrogels, skin regeneration

## Abstract

Melatonin and sericin exhibit antioxidant properties and may be useful in topical wound healing patches by maintaining redox balance, cell integrity, and regulating the inflammatory response. In human skin, melatonin suppresses damage caused by ultraviolet radiation (UVR) which involves numerous mechanisms associated with reactive oxygen species/reactive nitrogen species (ROS/RNS) generation and enhancing apoptosis. Sericin is a protein mainly composed of glycine, serine, aspartic acid, and threonine amino acids removed from the silkworm cocoon (particularly *Bombyx mori* and other species). It is of interest because of its biodegradability, anti-oxidative, and anti-bacterial properties. Sericin inhibits tyrosinase activity and promotes cell proliferation that can be supportive and useful in melanoma treatment. In recent years, wound healing patches containing sericin and melatonin individually have attracted significant attention by the scientific community. In this review, we summarize the state of innovation of such patches during 2021–2023. To date, melatonin/sericin-polymer patches for application in post-operational wound healing treatment has been only sparingly investigated and it is an imperative to consider these materials as a promising approach targeting for skin tissue engineering or regenerative dermatology.

## 1. Introduction

The skin with subcutis constitutes the largest organ in the human body, exposed to external and internal aging factors. The stochastic process of skin aging implies functional and phenotypic variability in cutaneous and immune cells, which occurs along with functional and structural changes in extracellular matrix constituents, including collagen and elastin. Wrinkling, roughness, skin laxity, and decrease in elasticity are the main clinical features of the skin aging process [1,2,3].

The crucial external factor of skin aging is the excessive exposure to ultraviolet radiation (UVR) which is connected with hyperpigmentary changes [4] and wrinkling [5], and also induces common types of skin cancer such as basal cell carcinoma [6,7], squamous cell carcinoma [8,9,10], and malignant melanoma [11,12]. The UVR spectra are: UVC (200–280 nm), UVB (280–315 nm) and UVA (315–400 nm), and radiation [13,14,15]. The UVC and 280–290 wavelengths of UVB are absorbed by the ozone layer of the atmosphere; however, the *stratum corneum* is able to absorb the UVC radiation after the exposition to non-natural light sources [14]. The UVA can penetrate the reticular dermis and induce biological effects, yet not as efficiently as the UVB radiation [16], which is absorbed by the upper layers of the epidermis and can penetrate the papillary dermis [13,15]. The UVR has an influence on many complex processes in the human body and can also trigger systemic reactions [17,18]. It can upregulate local and systemic neuroendocrine systems [18]. The locally induced cytokines, urocortins, enkephalins, and melanocortins can impose systemic effects while released into the circulation, like the agitation of the central hypothalamic–pituitary–adrenal axis, opioidogenic effects, and immunosuppression, as well as vitamin D synthesis and activation [17,18,19,20,21,22].

The UV lights have a tremendous impact on biological organisms and the origin of life on Earth [23,24,25,26]. The energy of UVR is similar to the energy of covalent bonds, which means that any molecule electron excitation by UV light can disrupt or create a covalent bond. Organic molecules exposed to the energy of UV light commute to chemical bonds with high-energy levels, achieving molecular complexity [18,26]. The cells can use their energy increasing the enthalpy of the system [26]. UVB is crucial in photosynthesis, providing various forms of vitamin D which can be enzymatically activated at the local and systemic levels in different organisms [21,22,27].

In human skin, reactive oxygen species/reactive nitrogen species (ROS/RNS) production can be induced by UVR, thereby elevating the secretion of pro-inflammatory cytokines. Increased cell proliferation and oxidation processes can disrupt cell membrane integrity and induce DNA fragmentation [7]. The topical delivery of antioxidants helps to maintain redox balance in epidermal cells, regulates pro-inflammatory cytokines release, and prevents oxidative damage and DNA fragmentation in these cells [28,29,30,31,32,33,34,35].

Melatonin (*N*-acetyl-5-methoxyindolamine) and sericin exert antioxidant properties and could be useful in topical wound healing patches by maintaining redox balance, cell integrity, and regulating the inflammatory response [36,37,38,39]. In human skin, melatonin suppresses the damage caused by UVR through numerous mechanisms associated with ROS/RNS production and programmed cell death (apoptosis) [40,41,42,43]. Melatonin derived from tryptophan constitutes an active pleiotropic molecule in the human organism, synthetized by cells of the pineal gland and also peripheral organs such as the skin, gastrointestinal tract, and lymphocytes [44,45,46]. Melatonin influences biorhythms, scavenges free radicals, enhances DNA repair, influences the gene expression of anti-oxidative enzymes, and stimulates wound healing properties [47,48,49,50,51]. Sericin as an effective antioxidant exhibits skin protective activity against UVB and UVA radiation-induced damage [52]. Chromophores’ UVB absorption is predominant, whereas UVA is weakly absorbed by DNA and cellular chromophores with greater ROS/RNS generation, which leads to oxidative changes in the cells [13,53]. Sericin is produced by silkworm’s glands (e.g., *Bombyx mori*, *Bombyx mandarins*, and other species) and has been explored in biomaterial applications because of its biodegradability, anti-oxidative, and antibacterial properties [54,55,56,57,58,59]. Furthermore, sericin inhibits tyrosinase activity and promotes cell proliferation, which can be supportive and useful in melanoma treatment [60,61,62,63,64]. After surgery, the potential bacterial infections could exacerbate epidermal/dermal damage and delay wound healing. Thus, developing biomaterials for melanoma therapy that reduces the risk of infection, maintaining redox balance in skin cells and preventing cancer recurrence is essential for patient recovery [65,66,67,68,69,70]. Sericin as a biomaterial has been used in the preparation of a variety of tissue engineering materials such as composites, hydrogels, membranes, nanofibers, and nanoparticles [71,72,73,74,75]. However, poor mechanical strength and high production costs of sericin-based scaffolds make it difficult to introduce this molecule for medical use. Using a biopolymer with improved mechanical properties and with the incorporation sericin and melatonin as active compounds may be highly useful for wound healing in melanoma-affected patients [76,77,78,79,80].

## 2. Melatonin and Sericin against the Skin Aging Process

The process of aging has a prominent impact on the skin’s healing function, mainly by prolonging the inflammatory phase and increasing ROS/RNS production [81,82]. The skin constitutes a protective barrier between external and internal environments and has the sensory capacity to maintain body homeostasis in response to deleterious factors. Skin aging is a natural process with progressive functional and morphological changes, determined by the overall exposure to both intrinsic and extrinsic factors, which may vary depending on skin regions within diverse ethnicities. The clinical signs of skin aging are visible as wrinkles, a rough-textured appearance, a loss of elasticity, and laxity [83,84].

The physiological process of skin aging is characterized primarily by fluctuation changes in endocrine circadian rhythmicity, gene expression, and hormonal descent, which are the reasons for appearance of morphological and functional alterations [85,86,87,88]. With increasing age, the occurrence of changes in the endocrine glands, the steroidogenic system, skin cholesterol synthesis, proopiomelanocortin (POMC) expression, and POMC-derived peptides production with the focus on the melanocortin receptor 1 (MC1R) and MC2R agonists, are more frequent and may lead to skin alterations and lesions [89,90,91,92]. Vitamin D production, essential to maintain proper skin functions and immunity, also decreases with age [92,93,94,95].

On the molecular level, the process of skin aging comprises ROS/RNS generation, diminished antioxidant protection, changes in gene expression, and defects in cellular DNA mechanisms. Along with the senescence of the organism, the mitochondrial DNA content and number decreases [96,97,98,99,100], but there is also enhanced ROS/RNS generation with reduced oxidative phosphorylation and adenosine triphosphate production which leads to mitochondria-mediated apoptosis [101,102,103]. Melatonin has an antioxidant capacity which relies on the indirect receptor-mediated stimulation of antioxidant enzymes to resist the oxidative stress [104,105,106,107,108,109,110,111,112]. Melatonin and its metabolites are also known for their anti-inflammatory and mitochondrial protective capability [32,113,114,115,116,117,118,119,120], which help to maintain proper skin functions [121,122,123]. Melatonin and its metabolites have a major role in human epidermal keratinocytes protection against UVB radiation, in particular *N*^1^-acetyl-*N*^2^-formyl-5-methoxykynuramine (AFMK), *N*^1^-acetyl-5-methoxykynuramine (AMK), 5-methoxytryptamine (5-MT), and 6-hydroxymelatonin (6(OH)MEL), which ameliorate the disruptive effects of UVR. Studies have shown that melatonin also protects the dermal fibroblasts from the deleterious action of UVA and UVB [124,125,126,127,128,129,130,131,132,133,134,135,136,137]. Thus, melatonin and its metabolites counteract photodamage and premature skin aging [138,139,140] (Figure 1). Finally, due to stimulated expression of involucrin, keratin-10 and keratin-14, topically applied melatonin enhances the epidermal barrier function of the skin and increases the activity of keratinocytes ex vivo [141,142] (Figure 1). The mechanism of action of melatonin and its metabolites would include the activation of the membrane-bound MT1 and T2 receptors [5] or the recently identified aryl hydrocarbon (AhR) and peroxisome proliferator-activated receptor gamma (PPAR-γ) [143], or the receptor independent mechanisms mentioned above.

Studies using dermal fibroblasts have shown that silk sericin stimulates collagen synthesis, which also indicates its anti-aging properties [144,145]. Results have revealed that silk sericin activates collagen type I synthesis and suppresses oxidative stress, maintaining unaltered fibroblast growth kinetics and cellular structure [146]. Next to ROS/RNS-scavenging activity, sericin also exhibits anti-tyrosinase and anti-elastase properties. Recent studies have shown that particular sericin strains have an anti-proliferative activity on peripheral blood mononuclear cells; in vitro IFN-γ secretion was decreased, without affecting TNF and IL-10 release. Thus, sericin may be useful for dermatological use [62,147].

The recent inventions using sericin in anti-aging treatments include extract loaded-sericin hydrogel as a topical agent [148], naringenin microemulsion-loaded sericin gel [149], and gold silk sericin/niacinamide/signaline complex [150]. Extract-loaded sericin hydrogels in six formulations were examined for anti-melanogenesis on the B16F10 melanoma cell line, UVR-preventive properties of human keratinocytes (HaCaT), and anti-aging effectiveness on normal human dermal fibroblasts. The study showed that the hydrogel increased the anthocyanin penetration through the skin. The most promising formulation using the purple waxy corn cob (*Zea mays* L.) extract demonstrated the highest tyrosinase activity inhibition, melanin pigment reduction, collagenase/elastase inhibition, collagen type I production, and elevation of cell viability.

Thus, the purple waxy corn cob extract-loaded sericin hydrogel as a topical agent indicates a great potential in anti-aging products [148]. Namely, naringenin microemulsion-loaded sericin gel showed an inhibition of UVR-induced photoaging and increased free radical scavenging. The in vitro cytotoxicity study on skin cancer cells enhanced anti-proliferative activity by increasing ROS/RNS in cancer cells with caspase-3 (Casp-3) activation [149]. The randomized study with the gold silk sericin/niacinamide/signaline complex have shown the efficacy of daily application in improving the condition of the skin with an antiaging effect [150].

UV light constitutes an important factor in skin aging and disturbances in skin proliferation. UV light up-regulates the nuclear factor kappa B (NF-κB) and releases pro-inflammatory cytokines, with a simultaneous increased generation of ROS/RNS. Next, free radicals affecting DNA decrease protein tyrosine phosphates and up-regulate matrix metalloproteinase generation, which leads to collagen decomposition [151,152]. Although, in healthy conditions urocanic acid, produced in the upper layers in the human skin, constitutes a natural skin UV absorber, excessive exposure to UV light should be avoided due to the harmful effects induced by UV light [153].

## 3. Melanoma: A Tumor of Melanocyte Origin

Melanoma, a tumor of melanocyte origin, constitutes one of the most formidable types of skin malignancy [154,155,156] described by local invasiveness, recurrence, early metastasis, and high mortality risk [157,158,159,160,161]. The standard treatment is surgical resection [162,163,164], while alternative therapies like chemotherapy, radiotherapy, or photodynamic therapy are focused on the elimination of the melanoma cells [165,166]. Nevertheless, the limitations of the treatment can cause prolonged stress for patients. Insufficient light penetration depth using photodynamic therapy may be a barrier for reaching pigmented lesions [167]. Furthermore, the complete tumor resection with residual tumor tissues may lead to major cutaneous defects [168,169]. The risk of the wound infection in post-surgical treatment is an emerging issue in the wound healing process. Therefore, it is essential to promote skin regeneration during melanoma treatment [170]. Throughout the years, various methods like autologous/allogenic skin grafts or tissue-engineered scaffolds were developed [171,172,173]. However, wound healing patches have drawn attention; understanding the requirements of melanoma treatment are indispensable to develop a strategy with integrated wound healing and therapeutic effects.

To better understand the process of melanoma wound healing, it is important to focus on the molecular bases of tumor development and possible ways to impede them [174,175,176]. Namely, the risk factors of melanoma induction are: UVR, burns, melanocytic nevi, fair skin or gene mutations (BRAF, NRAS, KIT) [177,178,179,180]. Also in the focus of attention is a tumor suppressor gene cyclin-dependent kinase inhibitor 2A (CDKN2A), and the encoding of p16INK4A and p14ARF proteins as a major genetic risk factor [181,182]. In the cell cycle of p16INK4α, the transition from G1 to S phase is regulated and the cycle also acts as a CDK inhibitor, blocking the phosphorylation and inactivation of the Rb protein. However, p14ARF shows anti-proliferation activity, inhibiting the disintegration of the p53 tumor suppressor. During the G2/M phase in normal conditions, the expression of p53 stops the cell cycle or induces apoptosis, which can be a response, for example, to UVR-induced DNA damage [183].

This process can activate the heat shock proteins (HSPs) which protect the cells from physical or environmental stressors [184]. In vitro studies have shown that after the exposure to UV light, melanoma cells release the HSP70 which activates anti-melanoma T cells [185,186,187,188]. Thus, HSPs play an essential role as molecular chaperones by assisting the correct folding of nascent and stress-accumulated misfolded proteins, and by preventing their aggregation [189]. Additionally, HSPs have a protective function; they allow the cells to survive in otherwise lethal conditions. Various mechanisms have been proposed to account for the cytoprotective functions of HSPs. Namely, several of these proteins have been demonstrated to directly interact with components of the cell signaling pathways, for instance those of the tightly regulated caspase-dependent programmed cell death machinery, upstream, downstream and at the mitochondrial level. HSPs can also affect caspase-independent apoptosis-like processes by interacting with apoptogenic factors, such as the apoptosis-inducing factor (AIF) or by acting at the lysosome level [189]. Melatonin is known to downregulate the expression of HSP40, HSP70, and HSP90 [32,190,191] to reduce oxidative stress. Thus, melatonin may constitute the additional defense in melanoma [32,192,193,194,195]. The anti-inflammatory properties of melatonin could also be an answer to the NF-κB activation, which is known to be the regulator in oncogenesis. NF-κB activation promotes cell proliferation and inhibits apoptosis and p53 in cancer development. The anti-inflammatory mechanism of melatonin’ purpose is to regulate cyclooxygenase-2 (COX-2), inducible nitric oxide synthase (iNOS), and cytokines [196].

Recent studies have confirmed that melatonin and its indolic and kynurenic derivatives downstream the pathway of melanogenesis, causing a drop in the cyclic adenosine monophosphate (cAMP) level and the microphthalmia-associated transcription factor (MITF) and causing the resultant collapse in tyrosinase (TYR) activity and melanin content. These findings can be a breakthrough for the future studies on pigmentation in melanoma therapies [122,142,196,197,198,199]. Since the inhibition of melanogenesis in advanced melanomas can serve as an adjuvant strategy in the systemic therapy of melanoma [200], melatonin as a mitochondrial protector with anti-inflammatory properties and direct antioxidants should be explored in topical and transepidermal delivery, especially in skin damage after UVR exposition. It is also known to be very effective in alopecia and atopic dermatitis treatment. It may be also considered as a skin barrier protectant in chronopharmacology [42,45,130]. In topical delivery, melanoma treatment using a combination of melatonin/sericin also indicates antioxidant and mitochondrial protection activity which may constitute a potential therapeutic strategy, including in the antibacterial properties of sericin that promote wound healing processes [201,202]. We also acknowledge the potential limitation of the accumulation of melatonin in tumor cells, which may increase their resistance to pharmaco- or radiotherapy, as discussed recently [203].

## 4. State of Innovation in Melatonin-Polymer Wound Healing Patches

Melatonin, a derivative of tryptophan, was initially characterized and isolated by Lerner et al. [204]. Melatonin is an indoleamine, which is defined as a indole heterocycle that contains two chains, i.e., 5-methoxy and 3-amide group. The indole structure is rich in electrons and has high resonance stability and reactivity which is the reason for melatonin’s free radical scavenging capacity [204]. To date, it has been reported that melatonin neutralizes a high number of ROS including hydrogenperoxide (H_2_O_2_), peroxyl radicals (ROO•), hydroxyl radical (•OH), singlet oxygen (^1^O_2_), and RNS, such as peroxynitrite (ONOO^−^) or nitric oxide radicals (NO•) [205,206,207,208,209,210,211]. The high antioxidant potential of melatonin enables the protection of the cells against oxidative stress more efficiently than other antioxidants. For instance, Figure 2 summarizes the cascade interaction between ROS/RNS and melatonin as well as its metabolites. Thus, melatonin has a capacity to detoxify numerous toxic ROS or RNS, whereas the scavenging ability of other antioxidants is significantly lower [212].

Furthermore, Table 1 presents melatonin as an active substance that is found in wound healing patches in combination with polymers: silk fibroin/methacrylate [213], polycaprolactone [214], carboxymethyl cellulose [215], methacrylated gelatin/thiolated pectin hydrogel [216], chitosan-sulfonated ethylene–propylene–diene terpolymer, sulfonated ethylene–propylene–diene terpolymer [217], nanoclay [218], polycaprolactone/sodium alginate [219], polycaprolactone/gelatin [220], collagen/chitosan cross-linked by glyoxal [221], gelatin [222], carboxymethyl chitosan [223], chitosan/collagen [224], chitosan–polycaprolactone/polyvinyl alcohol [225], chitosan–polycaprolactone [226], and collagen with aminated xanthan gum [227]. The main applications of melatonin-polymer patches are wound healing [214,223,224] and wound dressings [222,225], but they are also used in diabetic wound repair [215], skin delivery [218], and skin tissue engineering [221]. Furthermore, melatonin-loaded polymer patches were also used in cartilage repair [213], vital pump regeneration [216], tendon regeneration [219], membranes [217], nerve tissue engineering [220], skin tissue regeneration [227], and osteosarcoma treatment [226]. Polymers used in inventions were mainly synthetic but modified biopolymers including polycaprolactone [214], carboxymethyl cellulose [215], methacrylated gelatin/thiolated pectin hydrogel [216], chitosan-sulfonated ethylene–propylene–diene terpolymer [217], sulfonated ethylene–propylene–diene terpolymer [217], carboxymethyl chitosan [223], and chitosan–polycaprolactone [225]. A combination of natural and synthetic polymer were also observed, e.g., silk fibroin/methacrylate [213], polycaprolactone/sodium alginate [219], polycaprolactone/gelatin [220], collagen/chitosan cross-linked by glyoxal [221], chitosan/collagen [224], and collagen with aminated xanthan gum [227], although natural polymer matrix nanoclay was rare [218], as was gelatin alone [222]. Other active substances in wound healing patches found in recent years are *γ*-cyclodextrin [215], tideglusib [216], and silver nanoparticles [227].

## 5. State of Innovation in Sericin-Polymer Wound Healing Patches

Sericin is a “glue-like” protein coiled around the protein core which keeps the fibroin filaments together. This macromolecule constitutes a globular protein built from coil and *β*-sheets [228]. The coil structure for *β*-sheet can change in response to temperature, mechanical stretching properties, and moisture absorption. In 50–60 °C water solution, the protein acquires its soluble form. At lower temperatures, the coil structure converts into *β*-sheets resulting in the gel formation [229]. Sericin exhibits a hydrophilic character and is composed of 18 amino acids with polar groups such as hydroxyl, carboxyl, and amino groups. The group formation is capable of forming crosslinks, co-polymerizations, and compositions with other polymers [230]. The significant structure of sericin defines its biological properties that includes anti-bacterial activity, antioxidant, and biocompatibility [231].

The antioxidant and photoprotective potential of sericin against UVB in human epidermal keratinocytes was confirmed using the flow cytometry assessment [232]. It was revealed that treatment with sericin significantly attenuated apoptosis by inhibiting the expression of pro-apoptotic proteins and upregulation of the anti-apoptotic Bcl-2 proteins family, as well as preventing Casp-3 activation [232]. The role of sericin in preventing mitochondrial damage was also confirmed by the inhibition of hydrogen peroxide formation. As a consequence, the intracellular ROS/RNS and activation of poly-ADP-ribose polymerase enzyme (PARP) were distinctly reduced. Studies have shown that sericin is a potent antioxidant and anti-apoptotic agent [232,233]. Additionally, the antioxidant properties of sericin are related to high serine and threonine content, whereas hydroxyl groups act as a chelator of trace elements such as copper and iron [234,235].

In recent years, wound healing patches containing sericin have been the center of attention (Table 2). Sericin as a component of the matrix appeared next to polymers: poly (vinyl alcohol) [236], sodium carboxy-methyl-cellulose and polyvinyl alcohol [237], placenta-derived extracellular matrix [238], silk fiber [239], PVA/chitosan [240], PVA [241], robust alcohol, polyurethane/chitosan [242], carboxymethyl cellulose [243], poly (*N*-isopropylacrylamide) [244], cellulose/silk nonwoven fabric [245], collagen-fibrin [246], poly(2-hydroxyethyl methacrylate) [247], cellulose [248], carboxymethyl cellulose as hydrogels [249], cellulose/poly(vinyl alcohol) [250], poly lactide-co-glycolic acid [251], PVA/collagen [252], PVA/chitosan [253], polycaprolactone/cellulose acetate/fibroin [254], poly(ethyleneterephhalate)-*g*-poly(hydroxyethylmethacrylate, PET-*g*-HEMA) nanofibers [255], sericin/chitosan/polyvinyl alcohol [256], gelatin [257], fibroin [258], poly(Σ-caprolactone)/poly(ethylene oxide) [259]. There are also sericin-based patches with only sericin or sericin treated with HRP/H_2_O_2_ as a matrix, but in significantly smaller amounts [260,261].

The main active substances in those inventions included resveratrol [239], turmeric [243], silver nanoparticles [255], poly(lactic-co-glycolic acid) nanoparticles [257], and sericin as a sole ingredient [260,261] or in combinations with human placenta [238], silver [244], collagen [246] or azithromycin [250]. The applications of sericin-polymer patches in recent years involved wound treatment [238,239,243,244,245,249,250,252,254,257,259,261], including acute [237] and infected large burn wounds [236], artificial skin [242], wound dressings [240,241,248,253,255,258], tissue engineering [247], regenerative scaffolds [246], and periodontal tissue engineering [251]. The composition of the wound healing patches in the years 2021–2023 was mostly a combination of natural and synthetic components, where sericin was the natural ingredient. There were also fully natural patches designed from sericin and human placenta [238], as well as the combination of sericin and collagen [246].

The studies have shown that nanofiber (NF) wound dressing loaded with 20% melatonin (NF + 20% MEL) indicated the highest mRNA expression of collagen type 1 (COL1A1) on the 14th day of treatment compared to Comfeel Plus, NF + 10% MEL, nanofiber dressing, and the non-treated group. Thus, NF + 20% MEL electrospun wound dressing could be used as an effective matrix for accelerating the wound healing process reducing the costs of wound repair [225].

Another study has shown that the melatonin-loaded hydrogel significantly increased the percent of effectiveness of wound closure, promoted tissue granulation and re-epithelialization, leading to accelerated collagen deposition, when compared to the control group and the hydrogel with no melatonin groups. Therefore, it is essential to conduct further research using melatonin in wound healing patches [223].

## 6. Discussion: Melatonin/Sericin Collagen Scaffolds as Possible Wound Healing Patches

Recent advances in melatonin-polymer wound healing patches showed that melatonin is the main active ingredient [213,214,215,216,217,218,219,220,221,222,223,224,225,226,227]. Melatonin was also combined with other active substances to enhance regenerative, healing or anti-bacterial properties of dressings such as *γ*-cyclodextrin [215], tideglusib [216], and silver nanoparticles [227]. In comparison, sericin was used most often as a polymer in combinations with polyvinyl alcohol [237], carboxymethyl cellulose [243], sodium carboxymethylcellulose and polyvinyl alcohol [237], poly lactide-co-glycolic acid [251], PVA/chitosan [253], PVA/collagen [252], poly (*N*-isopropylacrylamide) [244], polyurethane/chitosan [242], poly(2-hydroxyethyl methacrylate) [247], polycaprolactone/cellulose acetate/fibroin [254], poly (ethylene terephthalate)-*g*-poly(hydroxyethylmethacrylate) (PET-*g*-HEMA) nanofibers [255], and poly(Σ-caprolactone)/poly(ethylene oxide) [259]. Less often occurring polymers used in combinations with sericin were gelatin [257], fibroin [258], cellulose [248], collagen/fibrin [246], placenta-derived extracellular matrix [238], silk fiber [239], and cellulose/silk nonwoven fabric [245]. Sericin used separately as a matrix was relatively rare, probably because of its mechanical properties [260,261]. Sericin used in the mentioned inventions were treated with HRP/H_2_O_2_ to enhance the mechanical properties [261]. When applied as a polymer of the matrix, it was used in combination with resveratrol [239], turmeric [243], silver nanoparticles [255], poly(lactic-co-glycolic acid) nanoparticles [257] as the active ingredients. Although sericin also constituted the main substance in the wound healing patches over the years 2021–2023 [236,237,238,239,240,241,242,243,244,245,246,247,248,249,250,251,252,253,254,255,256,257,258,259,260,261], it was effectively used in combinations with human placenta [238], silver [244], collagen [246] or azithromycin [250] to increase the efficacy of wound-healing dressings (Figure 3).

Polymers including melatonin were similar to those that were used in sericin wound healing patches such as polycaprolactone [214,259] and carboxymethyl cellulose [215,249], whereas similar polymers used in different combinations included silk fibroin/methacrylate, methacrylated gelatin/thiolated pectin hydrogel, poly(2-hydroxyethyl methacrylate) [213,216,247], chitosan-sulfonated ethylene–propylene–diene terpolymer, sulfonated ethylene–propylene–diene terpolymer [217], collagen/chitosan cross-linked by glyoxal [221], carboxymethyl chitosan [223], chitosan/collagen [224], PVA/sericin/chitosan [240,253,256], polycaprolactone/gelatin [220], gelatin [257], collagen with aminated xanthan gum [227] and collagen-fibrin [246]. Sericin-polymer patches in recent years have had a variety of applications as follows: wound treatment [238,239,243,244,245,249,250,252,254,256,257,259,261], including acute [237] and infected large burn wounds [236], artificial skin [242], wound dressings [240,241,248,253,255,258], tissue engineering [247], regenerative scaffolds [246], and periodontal tissue engineering [251]. The main applications of melatonin-polymer patches in the years 2021–2023 were wound healing [214,223,224] and wound dressings [222,225], also used in diabetic wound repair [215], skin delivery [218], and skin tissue engineering [221]. During this period, melatonin-polymer patches were employed in cartilage repair [213], vital pump regeneration [216], tendon regeneration [216], membranes [217], nerve tissue engineering [220], skin tissue regeneration [227], and osteosarcoma treatment [226]. Active compounds combined with resveratrol and turmeric exhibit antioxidant and anti-inflammatory properties [239,243], whereas silver nanoparticles or poly(lactic-co-glycolic acid) nanoparticles reveal anti-bacterial effect [255,257]. Thus, for wound healing patches it was essential to include them as proof of the regeneration processes of the skin. Finally, numerous reports have shown that both melatonin and sericin but also resveratrol and turmeric exhibit high antioxidant potential [262,263,264,265].

The elevating secretion of pro-inflammatory cytokines, increased cell proliferation and oxidation processes can damage cell membrane integrity and lead to DNA fragmentation [7,27]. Although UVR is one of the major melanoma risk factors it is also essential for initiating vitamin D synthesis, while active forms of vitamin D also have photoprotective properties [266,267,268]. The topical treatment of large or non-resectable melanoma lesions including lentigo maligna using a combination of melatonin and sericin also shows that mitochondrial and antioxidant protection activity may lead to the development of a valuable additional strategy targeting melanoma, regarding the antibacterial properties of sericin that accelerate wound healing [269,270].

## 7. Conclusions, Challenges and Future Perspectives

Sericin-polymer patches have a variety of applications such as wound treatment [237,238], including acute [237] and infected large burn wounds [236], artificial skin [242], wound dressings [240,241], tissue engineering [247], regenerative scaffolds [246], and periodontal tissue engineering [251], whereas the main applications of melatonin-polymer patches in years 2021–2023 were wound healing and wound dressings [222,223,224,225]; they were also used in diabetic wound repair [215], skin delivery [218], and skin tissue engineering [221]. During this period, melatonin-polymer patches were also used in cartilage repair, vital pump regeneration, tendon regeneration, membranes, nerve tissue engineering, skin tissue regeneration, and osteosarcoma treatment [213,216,217,220,226,227]. Research has shown that both melatonin and sericin exhibit high antioxidant potential that is beneficial in melanoma treatment [271]. Melatonin/sericin-polymer patches have not been exploited for their application in post-melanoma wound healing treatment; this is the field that would likely benefit from further investigation.

## Figures and Tables

**Figure 1 ijms-25-04858-f001:**
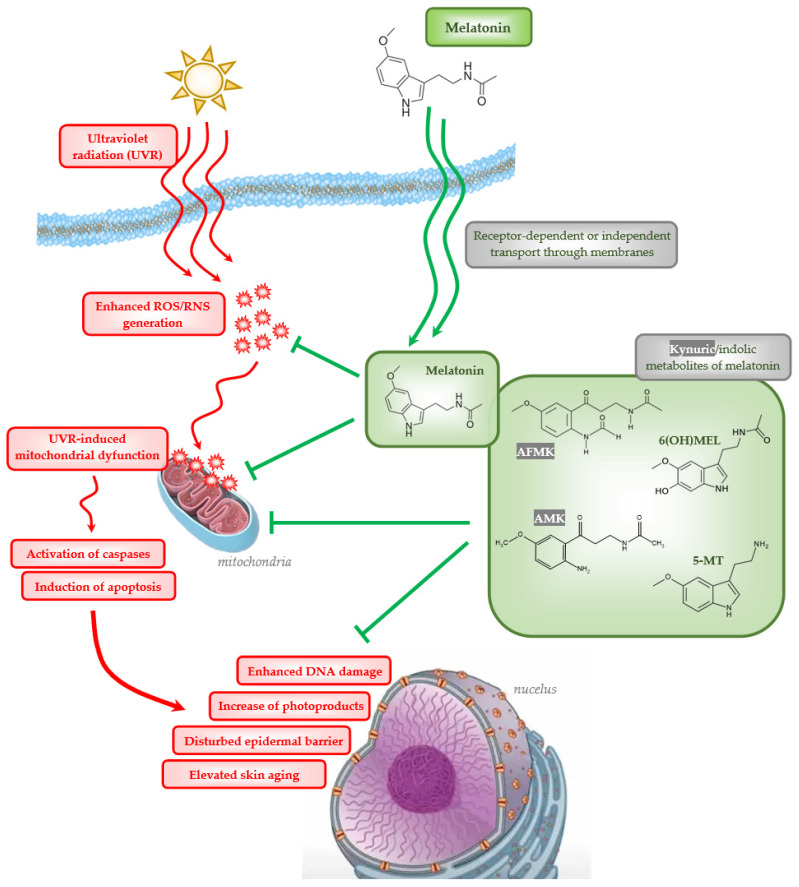
Cellular changes induced by ultraviolet radiation (UVR) including enhanced oxidative stress (ROS/RNS generation), mitochondrial dysfunction, DNA damage and protective action of melatonin as well as its kynuric (AFMK, AMK) and indolic (6(OH)MEL, 5-MT) metabolites.

**Figure 2 ijms-25-04858-f002:**
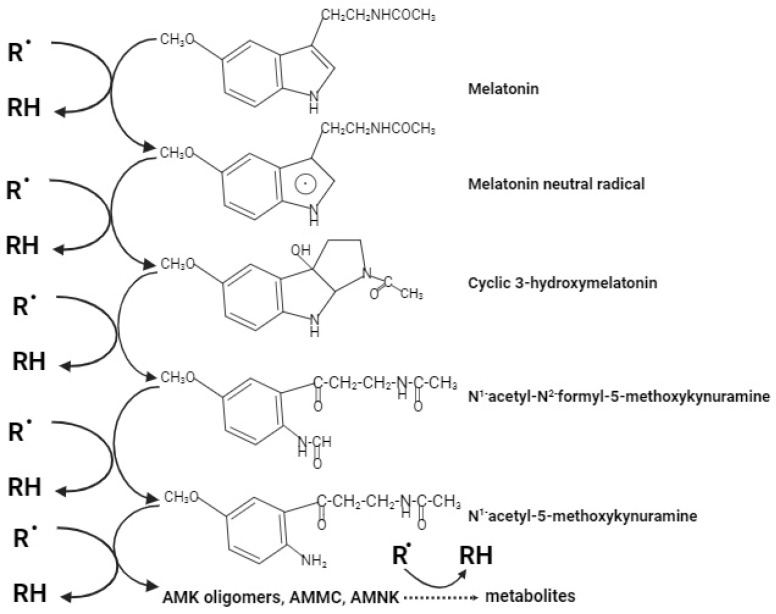
The cascade reaction with ROS/RNS products (or metabolites) of melatonin which are involved in terms of attenuation of UVR-induced changes as described above and presented in Figure 1.

**Figure 3 ijms-25-04858-f003:**
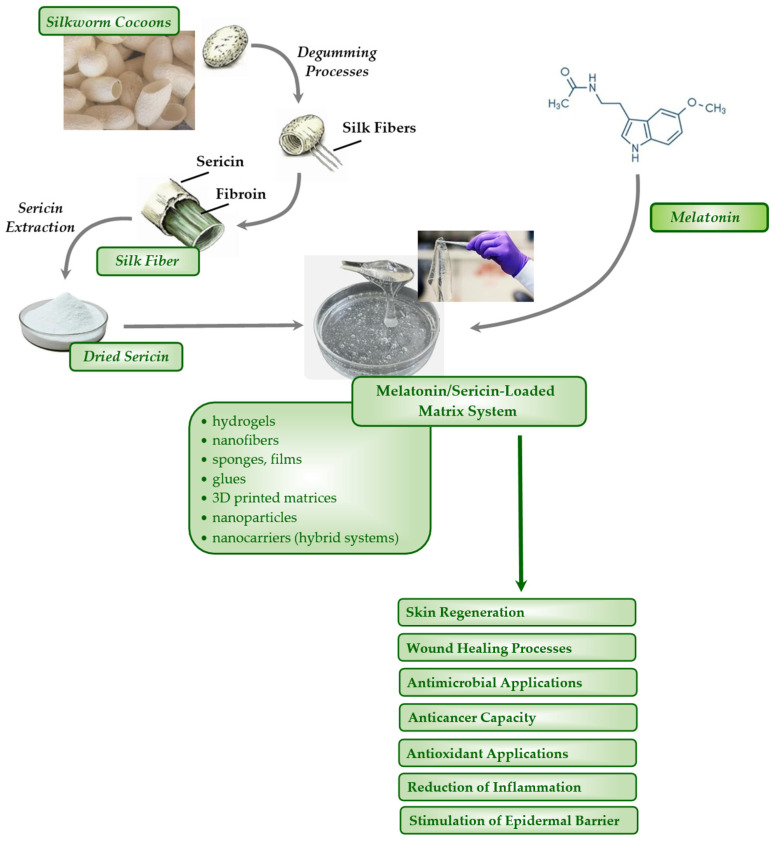
Melatonin/sericin-loaded matrix systems and their potential impact on skin-related applications.

**Table 1 ijms-25-04858-t001:** State of innovation in melatonin-polymer patches.

Invention	Matrix Polymer	Active Substance	Application	Origin	Year	Literature
Biomimetic melatonin-loaded silk fibroin/GelMA scaffolds	Silk fibroin/ gel methacrylate	Melatonin	Cartilage repair	Natural/ synthetic	2023	[213]
Melatonin-loaded polycaprolactone fiber mats	Polycaprolactone	Melatonin	Wound healing	Synthetic	2023	[214]
Carboxymethyl cellulose-based injectable hydrogel loaded with a composite of melatonin and *γ*-cyclodextrin	Carboxymethyl cellulose	Melatonin and *γ*-cyclodextrin	Diabetic wound repair	Synthetic	2023	[215]
Injectable methacrylated gelatin/thiolated pectin hydrogels carrying melatonin/tideglusib-loaded core/shell PMMA/silk fibroin electrospun fibers	Methacrylated gelatin/thiolated pectin hydrogels	Melatonin/ tideglusib	Vital pulp regeneration	Synthetic	2023	[216]
Chitosan–sEDMPand melatonin–chitosan–sEDMPcomposite membranes	Chitosan-sulfonated ethylene–propylene– diene terpolymer (Chi-sEPDM) membrane Sulfonated ethylene–propylene– diene terpolymer (sEPDM) membrane	Melatonin	Membranes	Synthetic	2023	[217]
Melatonin/nanoclay hybrids	Nanoclay	Melatonin	Skin delivery	Natural	2022	[218]
Biomimetic multilayer polycaprolactone/ sodium alginate hydrogel scaffolds loaded with melatonin	Polycaprolactone/ sodium alginate	Melatonin	Tendon regeneration	Natural/ synthetic	2022	[219]
Melatonin-polycaprolactone/ gelatin electrospun fibrous scaffolds	Polycaprolactone/ gelatin	Melatonin	Nerve tissue engineering	Natural/ synthetic	2022	[220]
Melatonin-cultured collagen/ chitosan scaffolds cross-linked by a glyoxal solution	Collagen/chitosan 3D scaffolds cross-linked by glyoxal	Melatonin	Skin tissue engineering	Natural/ synthetic	2022	[221]
Melatonin-loaded gelatin sponge	Gelatin sponge	Melatonin	Wound dressing	Natural	2022	[222]
Melatonin-loaded carboxymethyl chitosan (CMCS)-based hydrogel	Carboxymethyl chitosan (CMCS)-based hydrogel	Melatonin	Wound healing	Synthetic	2021	[223]
Polymeric matrix loaded with melatonin	Chitosan/collagen (CTS/Coll)-contained biomaterials	Melatonin	Wound healing	Natural/ synthetic	2021	[224]
Nanofiber wound dressing loaded with melatonin	Chitosan– polycaprolactone (PCL)/polyvinylalcohol (PVA)	Melatonin	Wound dressing	Synthetic	2021	[225]
3D-printing magnesium–polycaprolactone loaded with melatonin	Polycaprolactone	Melatonin	Osteosarcoma treatment	Synthetic	2021	[226]
Bio-hybrid hydrogel comprising collagen-capped silver nanoparticles and melatonin	Bio-hybrid hydrogel system comprising collagen and aminated xanthan gum	Silver nanoparticles and melatonin	Tissue regeneration in skin defects	Natural/ synthetic	2021	[227]

**Table 2 ijms-25-04858-t002:** State of innovation in sericin-polymer patches.

Invention	Matrix Polymer	Active Substance	Application	Origin	Year	Literature
Silk sericin/poly (vinyl alcohol) hydrogel	Poly (vinyl alcohol) hydrogel	Sericin	Infected large burn wound healing	Natural/synthetic	2023	[236]
Silk sericin-based hydrogel	Sodium carboxy-methyl- cellulose and polyvinylalcohol	Sericin	Acute wounds	Natural/ synthetic	2023	[237]
Sericin/human placenta-derived extracellular matrix scaffolds	Placenta-derived extracellular matrix	Sericin/ human placenta	Wound treatment	Natural	2023	[238]
Resveratrol loaded native silk fiber-sericin hydrogel	Hydrogel	Resveratrol	Wound healing	Natural	2023	[239]
PVA/sericin/ chitosan nanofibrous matrix	PVA/sericin/ chitosan	Sericin	Wound dressing	Natural/ synthetic	2023	[240]
Sericin/PVA hydrogels	Sericin/PVA hydrogels	Sericin	Wound dressing	Natural/ synthetic	2023	[241]
Robust alcohol soluble polyurethane/chitosan/ silk sericin (APU/CS/SS) nanofiber artificial skin	Robust alcohol soluble polyurethane/ chitosan/ silk sericin	N/A	Artificial skin	Natural/ synthetic	2023	[242]
Turmeric-loaded carboxymethyl cellulose/silk sericin dressings	Carboxymethyl cellulose/silk sericin	Turmeric	Wound healing	Natural/ synthetic	2023	[243]
Silver-loaded anti-bacterial sericin/poly (*N*-isopropylacrylamide) hydrogel	Sericin/poly (*N*-isopropylacrylamide) hydrogel	Silver/ sericin	Wound healing	Natural/ synthetic	2023	[244]
Cellulose/silk nonwoven fabric/silk sericin sandwich membrane	Cellulose/silk nonwoven fabric/silk sericin	Sericin	Wound healing	Natural/ synthetic	2023	[245]
Silk sericin- functionalized dense collagen/fibrin hybrid hydrogels	Collagen/fibrin	Sericin/ collagen	Regenerative scaffolds	Natural	2023	[246]
Sericin poly(2-hydroxyethyl methacrylate) hydrogel scaffolds	Poly(2-hydroxyethyl methacrylate)	Sericin	Tissue engineering	Natural/ synthetic	2023	[247]
Microstructured bacterial cellulose-silk sericin	Cellulose-silk sericin	Sericin	Wound dressing	Natural/ synthetic	2022	[248]
Carboxymethyl cellulose/sericin-based hydrogels	Carboxymethyl cellulose/sericin	Sericin	Wound healing	Natural/ synthetic	2022	[249]
Porous bacterial cellulose/poly(vinyl alcohol)-based silk sericin and azithromycin release system	Cellulose/ poly(vinyl alcohol)-based silk sericin	Sericin/ azithromycin	Wound healing	Natural/ synthetic	2022	[250]
Silk sericin/PLGA electrospun scaffolds	Silk sericin/ poly lactide- co-glycolic acid	Sericin	Periodontal tissue engineering	Natural/ synthetic	2022	[251]
Polyvinyl alcohol (PVA) hydrogel with collagen hydrolysate and silk sericin	PVA/collagen/ sericin	Collagen/ sericin	Wound healing	Natural/ synthetic	2022	[252]
PVA/sericin/ chitosan nanofibrous wound dressing matrix	PVA/sericin/ chitosan	Sericin	Wound dressing matrix	Natural/ synthetic	2022	[253]
Polycaprolactone/ cellulose acetate blended nanofiber mats containing sericin and fibroin for biomedical application	Polycaprolactone/cellulose acetate/sericin/fibroin	Sericin	Biomedical application	Natural/ synthetic	2022	[254]
PET-based nanofiber dressing material coated with silk sericin capped-silver nanoparticles	Poly (ethylene terephthalate)-*g*- poly (hydroxyethylmethacrylate) (PET-*g*-HEMA) nanofibers	Silver nanoparticles	Wound dressing	Natural/ synthetic	2021	[255]
Silver nanoparticles@organic frameworks/ graphene oxide (Ag@MOF–GO) in sericin/chitosan/ polyvinyl alcohol hydrogel	Sericin/chitosan/ polyvinyl alcohol	Silver nanoparticles	Wound healing	Natural/ synthetic	2021	[256]
Poly(lactic-co-glycolic acid) (PLGA) nanoparticles incorporated into sericin/gelatin nanofibers	Sericin/gelatin	Poly(lactic-co-glycolic acid) nanoparticles	Wound healing	Natural/ synthetic	2021	[257]
Silk sericin/ fibroin electrospinning dressings	Silk sericin/ fibroin	Sericin	Wound dressing	Natural/ synthetic	2021	[258]
Poly(*Σ*-caprolactone) poly(ethylene oxide) sandwich type nanofibers containing sericin-capped silver nanoparticles	Poly(*Σ*-caprolactone)/poly(ethylene oxide)	Silver nanoparticles	Wound healing	Natural/ synthetic	2021	[259]
Sericin scaffolds with ethanol post-treatments	Sericin	Sericin	Tissue engineering	Natural/ synthetic	2023	[260]
Horseradish perozidase- mediated cross-linked sericin hydrogels	Sericin treated HRP/H_2_O_2_	Sericin	Wound healing	Natural/ synthetic	2021	[261]

## Data Availability

Not applicable.

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
