# Peer review of "Melatonin/Sericin Wound Healing Patches: Implications for Melanoma Therapy"

_ijms, 2024, doi:10.3390/ijms25094858_

Round 1
Reviewer 1 Report
Comments and Suggestions for Authors
The review wrote by Adamiak and collaborators summirezed the latest innovation of topical patches in melanoma wound healing treatment. In particular, the authors analyzed the antinflammatory and antioxidant properties of Melatonin and Sericin alone or in combination with each other or with different polymers in patches and the application of these invention in wound healing and tissue rigeneration.
minor revisions:
1 In the second paragraph, please write POMC in full
2 The paper could be enriched by a figure representing the signaling pathways and the main genes activated in ROS production and damage caused by UVR.
3 The incipit of the following sentence in not very clear “The keratinocytes origin, Hsp70 protein is the reason of the further reactions like apoptosis, activation of p53 and proapoptotic protein translation into mitochondria, and caspase activation (Casp‐3 or Casp‐9)”, the authors should explain better the concept of keratinocytes origin and also the role of melatonin on Hsp70 protein expression.
4 Lane 280-283 the authors could specify the assays carried out to show the antioxidant and photoprotective potential of sericin against UVB. The author also wrote “upregulation of Bcl-2 antibody, maybe they meant Bcl-2 protein, please correct it.
5 Lane 385 “phtoprotective” There is a typing error
Author Response
The authors express their gratitude for all pointed out remarks. Thus, we addressed all of them accordingly.
Briefly, here is an additional figure (Figure 1) presenting counteracting capacity of melatonin, its kynuric (AFMK, AMK) and indolic (6(OH)MEL, 5-MT) metabolites against UVR-induced changes at the mitochondrial, nuclear and skin aging-related disturbances. Moreover, we reformulated the paragraph describing HSPs where we added respective reference, therefore the entire reference list has been updated and the numbering has been also revised throughout the manuscript.
Reviewer 2 Report
Comments and Suggestions for Authors
The manuscript titled 'Melatonin/Sericin Wound Healing Patches: Their Potential in Melanoma Treatment' is both interesting and timely. The authors have provided a thorough discussion of wound-healing patches and their potential for use in treating melanoma. While the manuscript contains sufficient information, a minor revision is required to improve it.
1. Figures showing valuable and representative results should be added,
2. Challenges and future perspectives should be offered in a separate section.
Author Response
Authors’ response: We want to thank for the remarks. Additionally, we enriched the manuscript with the additional figure (Figure 1) where we depicted the protective action of melatonin as well as its respective metabolites against deleterious changed triggered by UVR.